# Educational Apps and Dog Behavioural Problem Prevention: Associations Between the Zigzag Dog-Training App and Behavioural Problems

**DOI:** 10.3390/ani15040520

**Published:** 2025-02-12

**Authors:** Tom Rowland, Luciana de Assis, Carolyn Menteith, Lorna Winter, Helen Zulch, Daniel S. Mills

**Affiliations:** 1Animal Behaviour, Cognition and Welfare Research Group, School of Natural Sciences, University of Lincoln, Lincoln LN6 7DL, UK; trowland@lincoln.ac.uk (T.R.); lassis@lincoln.ac.uk (L.d.A.); hzulch@lincoln.ac.uk (H.Z.); 2Dogtalk, Gloucestershire, UK; carolyn@dogtalk.co.uk; 3Zigzag Petcare Services Ltd., 123 Buckingham Palace Road, London SW1W 9SH, UK; lorna@zigzag.dog

**Keywords:** animal behaviour, animal welfare, app technology, behavioural medicine, clinical animal behaviour, companion animals, veterinary behavioural medicine

## Abstract

It is important that puppies receive a good start to life to mitigate the risks of problematic behaviours developing. To achieve this, owner education is vital, and electronic applications (apps) provide a way to increase owner accessibility to welfare-friendly educational material. We therefore assessed associations between use of an educational dog behaviour app (Zigzag Dog Training app) and the development of a range of problematic behaviours. We found that app use was associated with a reduction in the severity of a number of problematic behaviours, most notably less aggressive behaviour toward familiar people, house soiling, chewing, barking, escape behaviour, and noise fear. Many other behaviour associations were favourable, but the sample was too small to draw reliable conclusions for them. Although this study cannot demonstrate a direct causal effect, it provides the first evidence of the potential value of well-designed educational dog behaviour apps and supports the need for further investigation.

## 1. Introduction

Broadly speaking, animal behavioural problems can be viewed as any behaviour that poses a problem to the animal’s carer [1]. These may be classified in terms of normal behaviours or dysfunctional, abnormal, and/or pathological processes [1]. Regardless, dog behavioural problems are a significant welfare concern for both the dogs in question and their carers. Many behavioural problems relate to negatively valenced forms of emotional arousal [2,3] and therefore represent a welfare concern in their own right. However, they are also commonly reported reasons for relinquishment to shelters [4,5,6,7,8,9], return to shelter [7,9], or euthanasia [10,11,12]. Further, problems associated with aggressive behaviours also come with risks of physical injury to the recipient [13,14], which may be the carers themselves or unknown members of the public. As such, there is also a public health aspect associated with aggressive behaviours [15].

Dog behavioural problems are also a mental health and wellbeing concern for the carers of these dogs. Problematic behaviour is associated with poorer wellbeing [16] and often negative emotions, which can place many practical constraints on a carer’s life [17]. Furthermore, individuals relinquishing their dogs tend to report lower attachment to their dog [6]. As such, while the precise causal pathways have not been delineated, a weaker owner–dog relationship and the stress and burdens associated with managing problematic dog behaviour may all be factors influencing the probability of relinquishment or euthanasia. However, even if relinquishment or euthanasia is not an outcome, there are clearly still welfare compromises to both dogs and their owners if behavioural problems are present.

Given the above, it is clear that developing effective interventions to treat and/or manage behavioural problems and improve owner–dog relationships is important. However, reactive treatment (e.g., psychopharmacology and/or behavioural modification plans implemented once a problem has developed) comes with downsides, and in many other fields, such as major depression in humans, there is expert consensus that prevention should be prioritized [18]. Further, a recent dog owner survey reported that ~82% of respondents utilized at least one aversive dog training technique, which was associated with the presence of behavioural problems [19], as has been reported elsewhere [20]. While it is unclear whether aversive techniques are used in response to the development of a problem or are a direct cause of a problem (or both), there are significant welfare consequences to dogs of such techniques [21,22,23,24,25,26,27,28].

Given current scientific understanding, an important part of problematic behaviour prevention may be ensuring that dogs have the best possible developmental experience. While there are obvious considerations for the period while with the breeder [29], it is also important to educate the owners who care for the dog through the rest of their development [29], with most owners obtaining puppies before the end of their sensitive period, between 7 and 8 weeks of age [30]. For example, poor socialization during puppyhood has been associated with increased probability of both social fears [31] and non-social fears [32], aversive training methods are associated with behavioural problems and reduced welfare [19,20,21], and low exercise/walking frequency is associated with behavioural problems [19]. While attendance at puppy classes is sometimes reported as a protective effect, being associated with reduced behavioural problems [19,20,33,34,35,36,37], not every owner may have a readily available opportunity to take their puppy to classes in its early weeks with them. During this period, important interactions and learning experiences would have taken place. As such, it is useful to develop educational resources for owners that can be accessed at the earliest point in the owner’s care of their puppy in order to promote beneficial experiences and learning.

Given the increasing use of technology in a large variety of aspects of modern life, there is growing interest in the use of educational applications (henceforth apps) to convey educational material promoting appropriate puppy development in a timely and accessible manner. Further, it is possible that some owners who may not seek out professional help may be more likely to use a smart phone app. The use of apps in people is fairly well established; for example, in the human literature, there is some evidence that app use in children has learning benefits [38] and that it can be beneficial in the treatment of some mental health conditions [39]. Additionally, in the companion dog literature, there has been work on app development for identifying signs of stress in dogs [40], an app targeted at breeders and owners primarily around veterinary health management [41], and one paper reporting the design of an interactive app for dog training [42]. App use for a variety of companion dog uses is therefore an ongoing line of research; however, to the authors’ knowledge, there has been no published assessment of the efficacy of dog-training-related apps in the scientific literature. One recent paid content app that has been developed is the Zigzag dog training app (henceforth Zigzag) developed by Zigzag Pet Care Services Ltd. Zigzag app content was initially created by applied and accredited trainers and behaviourists and has been adapted based on user feedback and feedback from an advisory board composed of other accredited trainers and behaviourists. While an exhaustive breakdown of the app content is beyond the scope of this introduction, we provide a summary of the first four chapters’ content in the Appendix A. Broadly speaking, Zigzag is a dog behaviour and welfare owner education tool that approaches puppy raising with an emphasis on emotional and cognitive development, rather than just a series of obedience exercises. Further details can be found at https://zigzag.dog/ (accessed on 20 November 2024).

In the current study, we undertook an online cross-sectional survey which sought to investigate associations between the use of the Zigzag app and problematic behaviours.

## 2. Materials and Methods

### 2.1. Ethics

This study was reviewed and approved by the University of Lincoln Ethics Committee (UoL-9472, 023).

### 2.2. Design

An online cross-sectional survey was undertaken to ascertain associations between Zigzag app use and behavioural problems. The survey was launched on 16 November 2022. The report here relates to data retrieved on 19 August 2024 from a subsection of the survey, which relates to problematic behaviour. A full copy of the survey can be found in the Appendix A.

In addition to owner and dog demographic questions, respondents were asked about app use and activities performed with their dogs. They were also asked about their perception of the severity of different behavioural problems. In brief, the 21 behaviour problem questions were owner-rated items designed to cover a range of problems owners might experience with their puppies. They were all scored on a 4-point ordinal scale ranging between “No problem at all (behaviour is absent or not worthy of note)”, “Mild (causes disruption but can be lived with)”, “Medium (difficult and persistent problem but you are learning to live with it/work around it)”, and “Severe (persistent problem causing disruption to your normal life and your expectations)”.

### 2.3. Data Collection

The survey was developed and distributed online via Qualtrics from November 2022, with the data used in the current study being downloaded on 19 August 2024. The survey was advertised via the social media of the authors and Zigzag Pet Care Services Ltd. in the following ways: tile pop ups within the app, emailing app users, a section on the Zigzag website containing a link to the survey, and a £1000 promotional advertisement on Facebook, which ran during February 2024.

### 2.4. Data Preparation

All data preparation was conducted in the R programming language ([43]: version 4.3.3), and the code for data preparation and statistical analysis can be found on the Open Science Framework (https://osf.io/usmyf/, accessed on 13 December 2024). Prior to statistical analysis, entrants that did not answer whether they enrolled on Zigzag were excluded (*n* = 744), entrants with a Q repcaptchaScore < 0.5 were excluded (*n* = 7), and entrants that answered that they performed “no training at all” but also answered “yes” to performing any of the other options for training activities were excluded (*n* = 357).

To isolate use of the Zigzag app from other activities that could influence development of behavioural problems, we focused on a subset of the data that only included entrants who had either carried out no training at all and those who had used Zigzag but who had not used any other free or paid dog/puppy app or attended puppy classes or any other formal in-person training (but they may have attended puppy parties or performed other informal social activities). This allowed us to focus on a population which may not seek out professional advice but may use an online educational app. Our primary explanatory variable of interest was defined as the percentage completion of the first four chapters of the Zigzag app, which contained the primary educational material of interest for problem prevention. For each chapter, participants were able to respond whether they had “Not read”, “Done some but not all”, or “Completed all exercises”. As such, to assign 100% to someone who completed all exercises from the first four chapters, we assigned the following numerical scores to each of their responses to the first four chapter completion questions: “Not read” = 0, “Done some exercises but not all” = (1/4)/2 = 0.125, and “Completed all exercises” = 1/4 = 0.25. We then summed the four values for each chapter for each participant to derive their completion percentage. Participants in the “no training” group were assigned a score of 0%.

### 2.5. Statistical Analysis

All statistical analysis was implemented in the R programming language [43] (v4.3.3), and the code can be found at the previously supplied OSF link.

Demographic information of respondents and their dog are presented as descriptive statistics with frequency and percentages and summarised with medians ± inter-quartile range unless otherwise stated. All quantitative information is rounded to 2d.p, except *p* values, which are rounded to 3d.p.

All behavioural problem variables were 4-point ordinal items. As such, we fit proportional odds ordinal logistic regression models to each question using *clm* in the R package *ordinal* [44]. Model fit was evaluated by examining surrogate residuals [45] using the R *sure* package [46]. The primary explanatory variable of interest was percentage completion of the Zigzag app. However, we express this percentage within the unit interval [0,1], such that coefficients in the estimated models represent a change from 0% to 100%. In all models, we also controlled for puppy age (in months), sex, whether a health issue was present or not, and whether the puppy was obtained from a rescue/abandoned or not. Note that these variables were simply used as statistical controls and were not the primary target of inference, and as such, information about them is only included in the Appendix A. The sample sizes for each model varied slightly, depending on missingness, which was removed row-wise.

Our primary aim was to estimate effect sizes for the associations, and as such, we report the odds ratio ±95% confidence interval for the Zigzag percentage completion variable from each model. Full information on each model can be found in the Appendix A. Given that this is an exploratory observational study, we make no adjustments for multiple comparisons, since the impact of false negatives was judged to be greater than the problem of false positives given the stage of scientific enquiry [47]. Furthermore, we generally take a compatibility approach to statistical inference [48], whereby the 95% confidence interval can be called a compatibility interval, which shows a range of values which are compatible with the data (odds ratios that would have *p* > 0.05 if used as the test hypothesis), with reported results emphasizing the point estimate and lower and upper bounds. We also provide the *p* value for the test against the null hypothesis of zero association. In compatibility terms, the *p* value is a measure of “the compatibility between the observed data and a targeted test hypothesis H, given a set of background assumptions” [48]. As such, the smaller the *p* value, the less compatible the data are with the test hypothesis (under the set of assumptions used to compute *p*). For select results, we use the *ggeffects* R package [49] to estimate the model-predicted probabilities marginalizing over the control variables, such that the plots show the predicted probability of being in each response category at different levels of Zigzag completion for an ‘average’ observation in the data [50].

## 3. Results

### 3.1. Demographic Information

After data cleaning, we had 1706 entrants with dogs between the ages of 3 and 24 months. After subsetting the data to include only entrants who did no training at all or those who only used the Zigzag app (see methods), we were left with 367 entrants (median puppy age in months = 4 ± 6). Of these, 194 did no training with their puppy, and 173 utilised the Zigzag app to varying degrees of completion of the first four chapters. Demographic information of these 367 owners and puppies is provided in Table 1 and Table 2, respectively. Descriptive statistics on the prevalence of the severity of the behavioural problems are provided in Table 3.

Table 4 reports the top 30 most frequent breeds within the Zigzag and no-training groups. The remaining breeds not listed only contained two or less individuals. The full table is provided in the Appendix A.

### 3.2. Behavioural Problems

Overall, the majority of odds ratios (19 out of 21) were in favour of problem severity reducing as a function of Zigzag completion (OR < 1), with the remaining 2 behaviours’ odds ratios marginally above 1 (Figure 1). However, compatibility intervals were generally quite wide, indicating that the estimates are reasonably imprecise, and a range of odds ratios are compatible with the data. Based on the point estimates, the *p* value, and values contained within the interval estimates, the results can generally be considered to fall into four categories, which we describe in turn below.

The first category includes behaviours with OR < 1 with *p* values < 0.05, and whose compatibility intervals fall within a range of practically meaningful reductions in problematic behaviour severity (reduced odds of selecting a more severe category). This includes house soiling, familiar aggression, escaping and chewing. Odds ratios however can be difficult to interpret. A more intuitive way is to consider what the model predicts in terms of the probability of being in each response category as a function of Zigzag completion. The predicted probability plots for each response category for house.soiling, familiar.agg, escaping and chewing are shown in Figure 2. In summary, and of note, for all these four behaviours, the predicted probabilities for the “No problem at all” category increased as percentage completion increased.

The second category includes behaviours with OR < 1 with *p* < 0.05, but whose compatibility intervals indicate that values ranging from very large reduction in odds of selecting a more severe category to only marginal reductions are all compatible with the data. This category includes noise fear and barking. The predicted probability plots for each response category for these two behaviours are shown in Figure 3. Again, of note is the fact that, for both behaviours, the predicted probability of the “No problem at all” increases as percentage completion increases.

The third category represents behaviours with OR < 1 but with *p* > 0.05 and *p* < 0.1, whose compatibility interval spans values including very meaningfully large reductions in the odds to only a neglible difference (OR~1). This includes the behaviours resource aggression and conspecific aggression. We also show the predicted probability plots for these behaviours in Figure 4. Again, note the increasing predicted probability of the “No problem at all” category as a function of percentage completion.

The final category includes behaviours with OR~1 with *p* > 0.1 and whose compatibility intervals indicate values ranging from meaningful reductions in odds to negligible or meaningful increases in odds are all compatible with the data. This last category are the behaviours that we are least certain about having a reliable relationship with use of the app and includes all behaviours not mentioned thus far.

There are not any behaviours which we are confident have practically no association at all with Zigzag completion rate. That is, there are no behaviours whose compatibility interval is very narrow, spanning values marginally below and marginally above 1. Similarly, there are no behaviours that we are confident are associated with an increase in severity as a function of app completion. That is, there are no behaviours whose compatibility interval spans only meaningfully large odds ratios.

## 4. Discussion

In this cross-sectional survey, we investigated associations between Zigzag app use and dog behavioural problems. The primary finding was that higher completion of the first four chapters of the app was generally associated with lower odds of having a more severe problematic behaviour, most notably for familiar.aggression, house.soiling, chewing, barking, escape, and noise.fear.

It is clear that for many behaviours assessed, given the relatively small sample, the compatibility intervals covered a range of values indicating considerable uncertainty regarding the precise size of association. However, despite this, all point estimates in the current study favoured the Zigzag app, except two, chasing and digging, whose odds ratios were marginally above 1, and neither of which are specifically addressed in the first four chapters of the app. Furthermore, no behaviours had any reasonable evidence that the app was associated with increasing severity. At worst, there were several behaviours whose intervals spanned a relatively large range of values below and above an odds ratio of 1. However, there were still several behaviours with reasonable evidence against a null association, with meaningful effect sizes in the interval range. These included familiar.agg, house.soiling, escape, and chewing. Regarding chewing, it is difficult to know precisely why chewing was occurring in the current sample, given that chewing can be a normal exploratory behaviour or associated with some form of stress [51,52,53]. Nonetheless, it is often considered a problem by owners [54,55]. As such, it is worth noting that Zigzag app use was associated with a fairly reliable reduction in severity of this problem. Further work is required to understand whether this is related to better caregiver understanding of the behaviour, the provision of chew toys (as taught in the App), or generally better caregiver understanding of puppy needs leading to more relaxed puppies overall.

The reduction in familiar aggression severity is obviously an important finding, given that aggressive behaviour usually represents some form of negative emotional arousal [2], comes with risks of injury to owners [13,14], and has been associated with relinquishment [4,5,6,8,9] and euthanasia [10,12]. Regarding house soiling, although it is an expected behaviour in young puppies, if it is prolonged, it may cause the owner stress and interfere with their relationship with the dog. Further, it has been reported as a reason for relinquishment [56]. As such, it is another important behaviour which showed good evidence of being quite strongly negatively associated with increasing app use, in that the severity of the problem markedly decreased, with the probability of selecting “No problem at all” increasing. Finally, the severity of the problem of puppies trying to escape was another behaviour we could be more confident about a reduction in. Given that escape could result in fatal injuries if an owner lived near road traffic, and that many countries have some form of legislation focused on being in control of a dog (e.g., section III of the UK Dangerous Dogs Act 1991 [57]), it is obviously vital to minimize escape problems as much as possible.

There were also indications that Zigzag was associated with reduced severity of noise fear and barking, albeit the upper bounds of the interval did not reach unequivocally meaningful effect sizes. Noise sensitivities are common, with previous reports suggesting ~50% of owners report at least one sign of fear in response to loud noises [58]. While it is becoming increasingly apparent that some noise sensitivities are related to pain [59], there are still many stimulus-specific fear-related associations that could be affected by a lack of habituation during puppyhood. Sound habituation exercises are introduced within the first four chapters of Zigzag. As such, it seems reasonable to suppose that app users were more likely to implement these exercises and thus better habituate their puppies to sounds, thereby reducing the severity of noise fears reported. Finally, barking showed a similar pattern of evidence. While barking occurs for many reasons [60], it is generally considered by owners to be a nuisance [61]. As such, whether the apparent reduction in barking severity is because Zigzag user’s dogs are less likely to react to fearful stimuli, exciting stimuli, and/or frustrating stimuli or is related to owners having better understanding of managing unwanted behaviours in order to prevent them escalating is difficult to know, but the identified association with reduced noise fear, could at least in part explain this effect. Regardless of the reason, the reduction is likely to be perceived positively by the owners, and it is one less problematic behaviour that could interfere with the human–animal relationship.

It is worth noting the findings for resource aggression and conspecific aggression, given the significance of the consequences of these behaviours. The point estimates indicated that the app was associated with notable reductions in the severity of these problems, albeit the interval widths covered ranges of effects with large beneficial reductions in severity to essentially a practically insignificant very small increase. Predicting aggressive behaviour is notoriously difficult due to its multifactorial nature, with even some of the best statistical models explaining <10% of the variance in a population [20]. It is thus to be expected that any effect will have wide confidence intervals. The maximum likelihood estimate in our study is consistent with Zigzag reducing the severity of these behaviours, but the estimate is too imprecise to make definitive conclusions. However, another contributing factor to the uncertainty is the low incidence of these problems (see Table 3). For example, for conspecific aggression, after removing missing data, ~86% of the sample answered that there was no problem at all, ~11% answered that there was only a mild problem, with ~2% answering moderate, and ~1% answering severe. This is apparent with the very small predicted probabilities for the more severe categories in Figure 4. A very similar response pattern was seen for resource aggression. As such, a larger sample size is likely required to capture what is likely to be a smaller effect given that the majority of dogs in our sample did not show aggressive behaviours. Overall, given the findings for all the aggressive-behaviour-related variables, it seems likely that Zigzag may be associated with reducing a general tendency to express aggressive behaviour (via some undetermined mechanism, for example, such as alterations in human–dog interactions, development of inhibitory control, improved emotional regulation, etc.) and deserves further exploration in future research.

While there are reports of engagement and appeal in veterinary-health-management-related apps targeting breeders and owners [41] and reports of developing interactive apps for dog training [42], to our knowledge, no study yet has examined associations between educational app use and owner-reported behavioural problems. There are a variety of educational resources in the dog bite prevention literature, with research generally demonstrating improved knowledge and retention of such knowledge, but with open questions about whether this directly translates into reduced dog bites (see [62,63] for overviews of this literature). Another common form of owner education is via puppy classes, for which there are some observational study data concerning associations with class attendance and behavioural problems. Generally speaking, puppy classes are associated with reduced aggressive behaviours [33,34], reduced likelihood of relinquishment [37], increased trainability [33,35,36], and lower anxiety-related behaviour [33,35]. This is broadly consistent with the association estimates of the current work, supporting findings of a beneficial association of owner education on reducing the risk of developing behavioural problems. While we do not compare app use and in-person class support in the current work, the relative efficacy of each would, however, be an interesting comparison to make, ideally as some form of randomized trial. It is also worth considering why we observe associations for some behaviours and not others. Fundamentally, this may simply be a statistical issue in terms of low power and imprecise estimates. Our statistical estimates cannot exclude beneficial associations for any of the behaviours in the third and fourth ‘categories’ described in the Results Section. The only effect we have no evidence for is large detrimental associations with app use. As such, the current data do not necessarily support that the app only affects certain behaviours and not others. Rather, there are some which have more evidence of a beneficial association and others which, at the present time, we remain uncertain about. That being said, it is possible that the app is more effective for some types of behaviours than others. While speculative, and not a complete explanation for our pattern of results, it is possible that the app does a good job at improving owner–dog interactions, thereby reducing the probability of household aggressive behaviour and resource-related aggressive behaviour. Further, it may give simple advice for common puppy problems, such as house soiling and chewing, which owners find easy to implement with minimal effort. The same may be true for escaping, where simple advice on ensuring the property is secure is sufficient. However, other problems may involve more active training or effort to address. For example, structuring periods of separation from the dog, taking the dog to visit unfamiliar people, or lots of practice at differential reinforcements to address behaviours, such as food stealing, jumping, chasing, and digging (although digging is not specifically addressed in the first four chapters). Direct measures of app user implementation of advice would be useful here to tease apart these possibilities, a point we return to later in the limitations paragraph below.

One major advantage that apps may have over other forms of education within the context of dog training and behaviour is that they may be able to target a wider audience and establish engagement among a population who may not otherwise seek the help of professionals. While the current work was not designed to address whether this is the case or not, compared to Mead et al. [64], our sample contained more younger individuals and first-time dog owners. It is tempting to speculate that the app may attract young(er) first-time owners, although definitive conclusions would require a more detailed analysis of app user information. Relatedly, given that a number of apps have been developed and tested for young children’s learning (see [38] for overview), it would be interesting to examine further whether the use of apps is an effective way of increasing dog behaviour and training knowledge among children. In particular, whether they retain the knowledge and whether it reduces, for example, the incidence of dog bites toward children. Furthermore, exploring which demographic and person characteristics are associated with dog app use may help inform the design specifics of apps to maximise engagement. However, whether app use is a sufficient substitute for advice from an appropriately qualified professional is, at present, unknown. If this is not the case, then one would not want apps to reduce the probability of people seeking such help. As such, we feel the behaviour of dog-training-app users is also an important area to investigate within this research context. Of note, however, is that the Zigzag app comes with support from a team of professionals which users can contact, and approaches such as this may be a useful way to encourage app engagement and possibly even increase the probability of contacting professionals in a timely manner if it is easy to do so within the app.

A reduction in the severity of problematic behaviour comes with an obvious direct welfare benefit, in that negative emotional states associated with such behaviour will be reduced. However, it is also likely that by reducing the development or severity of problematic behaviours, there will be an indirect welfare benefit, in that it may reduce the use of aversives towards dogs [65]. Aversive training is generally associated with behavioural problems and reductions in welfare in the dog-specific literature (e.g., see [22] for a review in dogs), and the negatively valenced emotional states resulting from punishments are well defined in general animal welfare theory [25,26,27]. Further, recent estimates of the incidence of utilizing at least one form of aversive technique is still very high at ~82% [19]. Unfortunately, we do not have data on training approaches used by participants in the current study (other than knowing the Zigzag app promotes welfare-friendly approaches). As such, while this is speculation at present, if educational apps such as Zigzag are able to reduce the severity of problematic behaviour, this may in turn reduce the incidence of aversive training if owners do not feel they need to resort to such approaches because of an unmanaged problem. Further, if there is a causal effect of app use on reducing problematic behaviours, particularly those associated with relinquishment, then it would be interesting to perform longitudinal follow-up studies to assess whether app use is associated with lower rates of relinquishment and/or behavioural euthanasia.

The primary limitation of the current work is that it is observational, based on a cross-sectional sample, for which, confounding variables are likely to be present. For example, the major confound is that individuals who use the app may well be people who are likely to try to do the best for their dog and will naturally have a better relationship with their pets. We tried to somewhat account for this by focusing on a subset of the data that did not contain individuals who did multiple activities with their dog and including level of engagement with the app as a predictor. However, the limitation remains that no causal claims about efficacy can be established on the basis of the current work. However, there are reasonable logical causal arguments that might be made from the results, which deserve consideration. Ideally, a randomized clinical trial would be required to be able to make causal claims, or alternatively, causal inference methods could be used, but such an approach requires very strong assumptions and well-defined structural causal models [66]. Further, while we attempted to model engagement with the app by quantifying the chapter completion percentage, it should be noted that this is not a direct quantification from app data, and it is also not a quantification of what the owner actually implemented as advised by the app. Future work should consider measuring these aspects of engagement, which will likely strengthen the statistical estimates and the conclusions that can be drawn. One final consideration is the wording of the questionnaire items used in the present work. These were designed to capture what owner’s perceived as problematic, given that it is owners that present their dog for behavioural complaints which they perceive as problematic [1]. However, this should be taken into account when interpreting the results.

## 5. Conclusions

Increasing completion of the Zigzag app was generally associated with decreased odds of problematic behaviour severity. Our results are the first to indicate the potential value of dog-training apps (albeit limited to a single product). Future research should continue to investigate a range of aspects relevant to this research line of enquiry, including mode of delivery and engagement within the app, the relationship between app use and engagement with professionals, testing the retention of knowledge and implementation of app advice, estimating long-term outcomes, and finally, ideally implementing randomized trials to provide unbiased estimates of causal effects. Given the potential long-term impact of poor-quality advice to puppy owners on future behaviour, it is important that the quality of any widely used intervention on puppies is assessed. We suggest they have the potential to play a useful role for owners, but further research is warranted, and generalization between products is unwarranted.

## Figures and Tables

**Figure 1 animals-15-00520-f001:**
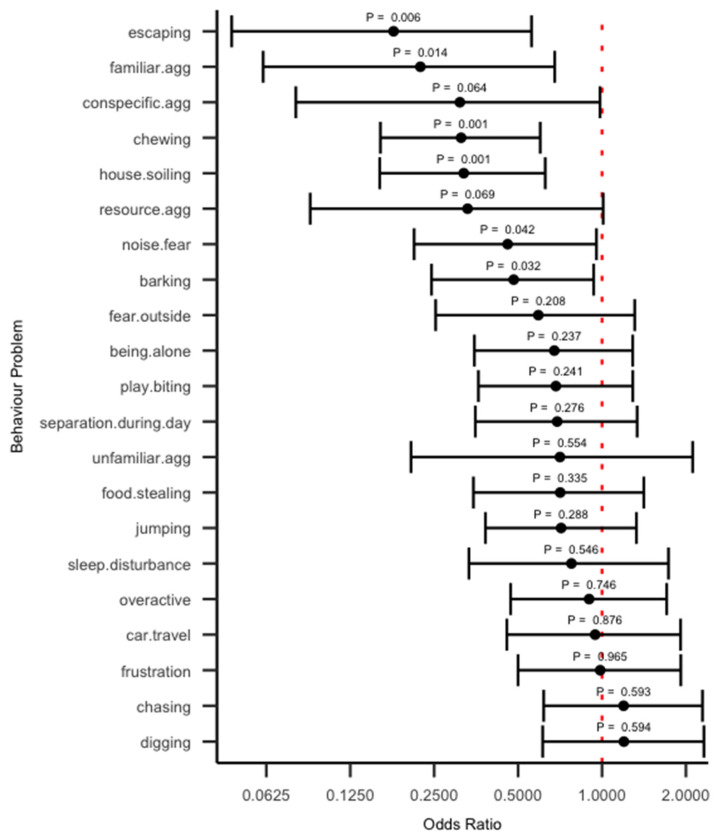
Associations between Zigzag completion and behavioural problems from the proportional odds ordinal logistic regression models. Dots represent the odds ratio point estimates with associated *p* values printed above these, and the error bars represent the 95% compatibility intervals. The vertical dotted red line indicates an OR = 1 which represents no association. Note that the x scale is on a log10 visual scale to ease visualisation. Behaviours are ordered by the odds ratio point estimate.

**Figure 2 animals-15-00520-f002:**
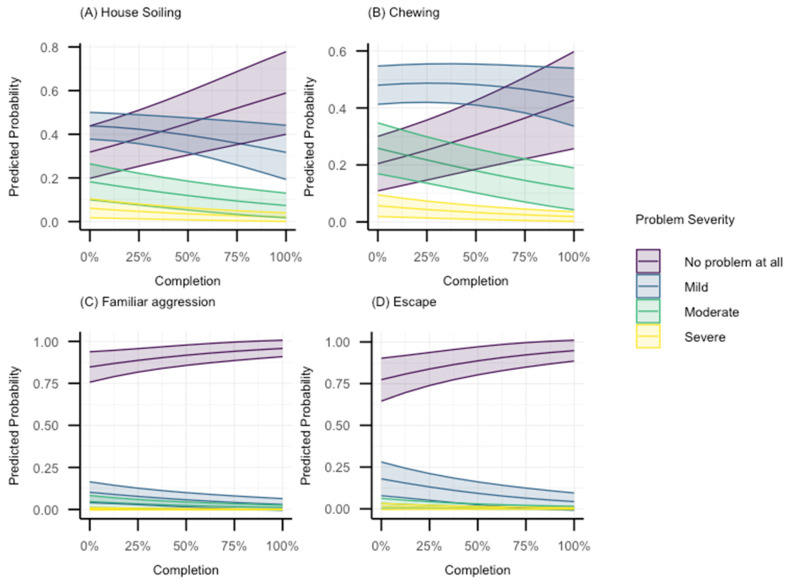
Model-predicted probabilities for each response category as a function of Zigzag completion for (**A**) house.soiling, (**B**) chewing, (**C**) familiar.agg, and (**D**) escape. Note that the y/predicted probability limits vary between plots.

**Figure 3 animals-15-00520-f003:**
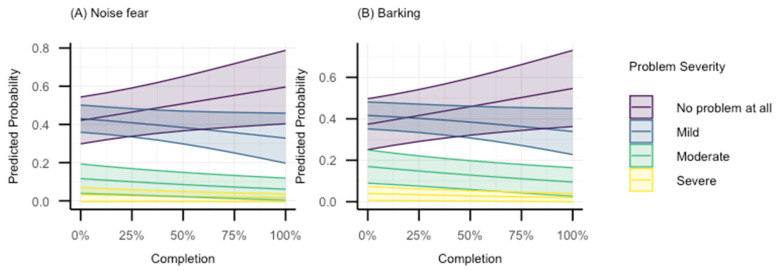
Model-predicted probabilities for each response category as a function of Zigzag completion for (**A**) noise fear and (**B**) barking.

**Figure 4 animals-15-00520-f004:**
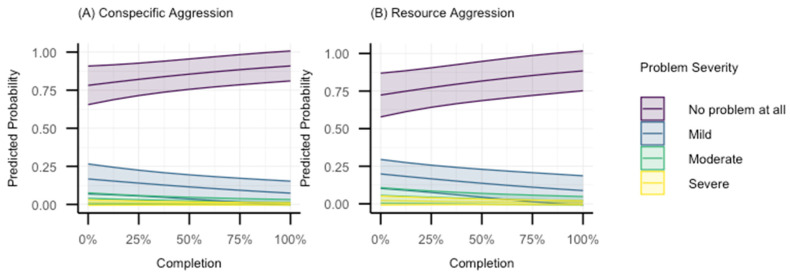
Model-predicted probabilities for each response category as a function of Zigzag completion for (**A**) conspecific.agg and (**B**) resource.agg.

**Table 1 animals-15-00520-t001:** Demographic information of puppy owners.

Variable		N (%)
Owner’s gender	Female	321 (87.5)
	Male	42 (11.4)
	Non-binary/third gender	3 (0.8)
	Prefer not to say	1 (0.3)
Owner’s age	18–25	41 (11.2)
Owner’s age	26–35	91 (24.8)
	36–45	62 (16.9)
	46–55	61 (16.6)
	56–65	70 (19.1)
	66+	42 (11.4)
Country	United Kingdom	230 (62.7)
Country	Australia	7 (1.9)
	New Zealand	2 (0.5)
	South Africa	4 (1.1)
	United States	92 (25.1)
	Other	31 (8.4)
	Missing	1 (0.3)
Owners that have had dogs before	Yes	260 (70.8)
Owners that have had dogs before	No	82 (22.3)
	Missing	25 (6.8)

**Table 2 animals-15-00520-t002:** Demographic information of the puppies.

Variables		N (%)
Where puppy was from	Bred myself	9 (2.5)
	Family/friend	48 (13.1)
	Found abandoned	6 (1.6)
	Private breeder	240 (65.4)
	Shelter/rescue centre	23 (6.3)
	Other	16 (4.4)
	Missing	25 (6.8)
Puppy’s sex and neuter status	Male entire	180 (49)
	Female entire	127 (34.6)
	Male neutered	21 (5.7)
	Female neutered	19 (5.2)
	Missing	20 (5.4)
Puppies with ongoing health issues	Yes	20 (5.4)
	No	322 (87.7)
	Missing	25 (6.8)
Period puppies are left alone on an average day	Never	103 (28.1)
	1–2 h	134 (36.5)
	2–4 h	68 (18.5)
	4–6 h	32 (8.7)
	6–8 h	3 (0.8)
	8–10 h	2 (0.5)
	10–12 h	0 (0)
	More than 12 h	0 (0)
	Missing	25 (6.8)

**Table 3 animals-15-00520-t003:** Responses to behaviour problem questions in the survey with the associated shortened term used in text and frequency of endorsement (percentages in brackets) of each response category for the total sample (*n* = 367) containing only no-training individuals (*n* = 194) and Zigzag users (*n* = 173).

Behaviour Problem Question	Short Term	“No Problem at All”	“Mild”	“Moderate”	“Severe”	Missing
Does your puppy have an issue with chewing?	Chewing	77 (21.0)	154 (42.0)	70 (19.1)	14 (3.8)	52 (14.2)
Does your puppy have an issue with play biting?	Play.biting	77 (21.0)	129 (35.2)	85 (23.2)	24 (6.5)	52 (14.2)
Does your puppy have an issue with house soiling?	House.soiling	137 (37.3)	118 (32.2)	45 (12.3)	15 (4.1)	52 (14.2)
Does your puppy have an issue with aggression (growling, snarling, snapping, or biting outside of play) towards human family members or close friends?	Familiar.agg	268 (73.0)	32 (8.7)	14 (3.8)	1 (0.3)	52 (14.2)
Does your puppy have an issue with aggression (growling, snarling, snapping, or biting outside of play)towards strangers?	Unfamiliar.agg	278 (75.8)	31 (8.5)	4 (1.1)	1 (0.3)	53 (14.4)
Does your puppy have an issue with aggression (growling, snarling, snapping, or biting outside of play) to other dogs?	Conspecific.agg	269 (73.3)	35 (9.5)	7 (1.9)	2 (0.5)	54 (14.7)
Does your puppy have an issue with jumping up?	Jumping	65 (17.7)	149 (40.6)	79 (21.5)	20 (5.5)	54 (14.7)
Does your puppy have an issue with car travel?	Car.travel	210 (57.2)	80 (21.8)	15 (4.1)	8 (2.2)	54 (14.7)
Does your puppy have an issue with chasing others or moving objects?	Chasing	177 (48.2)	99 (27.0)	31 (8.5)	6 (1.6)	54 (14.7)
Does your puppy have an issue with digging?	Digging	171 (46.5)	101 (27.5)	32 (8.7)	7 (1.9)	56 (15.3)
Does your puppy have an issue with being alone?	Being.alone	129 (35.2)	102 (27.8)	57 (15.5)	23 (6.3)	56 (15.3)
Does your puppy have an issue with noise fear?	Noise.fear	194 (52.9)	96 (26.2)	17 (4.6)	4 (1.1)	56 (15.3)
Does your puppy have an issue with barking?	Barking	148 (40.3)	118 (32.2)	37 (10.1)	8 (2.2)	56 (15.3)
Does your puppy have an issue with food stealing?	Food.stealing	196 (53.4)	75 (20.4)	30 (8.2)	10 (2.7)	56 (15.3)
Does your puppy have an issue with escaping?	Escaping	258 (70.3)	43 (11.7)	7 (1.9)	3 (0.8)	56 (15.3)
Do you have an issue with your puppy being overactive?	Overactive	141 (38.4)	128 (34.9)	35 (9.5)	7 (1.9)	56 (15.3)
Does your puppy have an issue with growling, snapping, or biting to protect something (like food, toys, bed, etc.)	Resource.agg	266 (72.5)	34 (9.3)	8 (2.2)	3 (0.8)	56 (15.3)
Does your puppy have an issue with separation-related problems during the day?	Separation.during.day	155 (42.2)	104 (28.3)	33 (9.0)	19 (5.2)	56 (15.3)
Does your puppy appear to freeze up in fear or avoid specific things while outside?	Fear.outside	224 (61.0)	63 (17.2)	15 (4.1)	5 (1.4)	60 (16.4)
Does your puppy have an issue with disturbing you while you sleep at night?	Sleep.disturbance	228 (62.1)	69 (18.8)	7 (1.9)	3 (0.8)	60 (16.4)
Does your puppy appear to have an issue with frustration, i.e., resisting being restrained or struggling to cope when they cannot access something that they want?	Frustration	175 (47.7)	95 (25.9)	30 (8.2)	7 (1.9)	60 (16.4)

**Table 4 animals-15-00520-t004:** Frequency of the top 30 breeds in the Zigzag and no-training groups. All other breeds had only two or less individuals in total.

Breed	No Training	Zigzag	Breed	No Training	Zigzag
Labrador Retriever	16	20	Staffordshire Bull Terrier	5	2
Cocker Spaniel	17	12	Crossbreed	2	4
Cockapoo	15	10	Whippet	2	4
Missing Breed Data	5	17	Goldendoodle	3	2
Mixed Breed	10	5	Minature Schnauzer	2	3
Border Collie	7	6	West Highland White Terrier	3	2
Cavapoo	6	3	Beagle	3	1
German Shepherd Dog	2	7	Chihuahua	2	2
Golden Retriever	4	5	Yorkshire Terrier	4	0
Siberian Husky	5	4	Fox Terrier Wire	0	4
Dachshund Minature Smooth Haired	4	4	American Pitbull Terrier	3	0
English Springer Spaniel	3	5	Cavapoochon	1	2
Other	4	4	Chow Chow	2	1
French Bulldog	6	1	Maltipoo	3	0
Labradoodle	3	4	Rottweiler	3	0
Shih Tzu	4	3			

## Data Availability

The final dataset for use in the current work can be found at the following Open Science Framework link: https://osf.io/usmyf/ (accessed on 1 December 2024).

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
