# Peer review of "Educational Apps and Dog Behavioural Problem Prevention: Associations Between the Zigzag Dog-Training App and Behavioural Problems"

_animals, 2025, doi:10.3390/ani15040520_

Round 1
Reviewer 1 Report
Comments and Suggestions for Authors
Title: Educational Apps and Dog Behaviour Problem Prevention: Associations Between the Zigzag Dog Training App and Behaviour Problems
Summary:
This paper outlines an early investigation of the likelihood of owners reporting problem behaviors in their puppies based on their use of a dog training phone dog called Zigzag. They identify a need for research on prevention and early intervention before problem behaviors have serious consequences. The authors analyzed a subset of data from a much larger pool so that they were specifically analyzing data from owners who did no formal training or only used the Zigzag education app. To varying degrees, use of the app was associated with a reduced likelihood of reporting problem behaviors. Although the study was observational in nature, it is a first step in identifying several new directions for future studies.
General concept comments:
The topic of this manuscript is of great value. The authors identify that problem behaviors in domesticated dogs often result in relinquishment of pets to shelter and/or euthanasia. Owner education is an important factor in solving problem behaviors and this study investigates a new way of disseminating that information, a phone app Zigzag. This could be an important step to broader education of the public early in the dog’s development as a prevention tool.
Only about half of the references are recent, within 5 years. There are some notable exceptions to the need for recent work. For example, sources about statistical analysis are justifiable in their use as once the seminal information is published, there is no need for additional sources. However, some of the sources about why owners relinquish pets are 20-30 years old as are many sources about problem behaviors in dogs. One suggestion would be to add recent sources. Training methods have evolved since the early 2000s and relying on outdated training resources does not strengthen the paper. Additional sources about aggressive behaviors in dogs are also necessary. Additional, recent, research on relinquishment statistics will also strengthen the author’s claim that there needs to be more emphasis on prevention.
Demographic information was collected and presented in tables 1 and 2. However, there is no later analysis of these data. For breeds, there are likely not enough dogs in some breeds to run appropriate analyses. Analyses showing whether owner or dog gender had any effect would strengthen the results section. Further, this is a group of owners who do very little, if any, training with their dogs, yet the demographic information reported shows that the majority of these dogs were purchased from private breeders. There seems to be a disconnect between how carefully a dog is chosen and how much time the owner spends training. This is worth mentioning in the discussion section.
The owners who used the Zigzag app were categorized as completing 1, 2, 3, or 4 chapters of training modules. While this is a helpful classification, it does not capture differences in engagement levels. The authors note that they were able to distinguish how much of each chapter was completed by the participant, but the inability to identify the quality of the engagement should be noted and explained in the discussion as a possible confound. Similarly, if all owners interacted with the app in exactly the same way, this should be noted within the methodology section for clarity.
The authors were careful to point out the limited scope of the results and proposed several future directions which were appropriate recommendations.
Specific Comments
Introduction lines 65-74, the authors discuss reactive treatment but do not provide further explanation or examples of reactive treatments.
Figure 1 caption: identify the red line at Odds Ratio = 1.000 in the caption.
Reviewer 2 Report
Comments and Suggestions for Authors
Overview
This paper investigates the use of an app to help educate dog owners in overcoming problem behaviours via surveying a population of owners that have only used the app and no other information sources to gain dog training and behavioural modification advice. Model predicted probabilities for various problem severity categories as a function of Zigzag (app) completion were presented alongside other descriptive data.
General Feedback
A novel area of dog science worth further exploration, this study sets about investigating how use of an app could be beneficial to those wishing to improve their dogs’ behaviours. This paper has great potential and makes a reasonable case for the value of app technology in providing dog guardians with training resources to improve their dogs’ behaviours. Overall, there are a few sections that could be developed to maximise the paper’s usefulness to this emerging field. App use in this area is still quite rare so even discussing the basic comparisons with other training platforms to users lends value to this subject area (see intro/lit suggestions/comments). A lot of data has been collected and I had a few extra questions around the topic whilst reading (see below). This is a valuable area of work and providing some of these additional answers will help ensure this paper’s continued value to future studies in this area.
You can see this work has taken huge effort and it has paid off. Further suggestions are provided below per section:
Abstract:
Suggestion to add the sample size to abstract since mentioned.
Intro:
Does “app” need to be referred to as “application” with abbreviation on first use?
Add a definition for behavioural problems. Could include more on how different types of PB are viewed or handled generally due to their great diversity. (Discussion point - Would any lend themselves more to app training over others and why?)
Suggest it may be useful to discuss/reference the use/success of apps for human behaviour change or knowledge uptake outside of a dog behaviour context to build greater context/justification for investigating this area e.g., Griffith et al 2020 Apps As Learning Tools: A Systematic Review. It could then be referred to in discussion also. Recommend expanding paragraph 5. Greater introduction to this type of technology and other study’s findings / pros and cons for in animal care could help set scene further? Some examples:
Choi, C. H., Park, S. W., Jung, S. H., & Sim, C. B. (2023). Implementation of a Mobile App for Companion Dog Training using AR and Hand Tracking. The Journal of the Korea institute of electronic communication sciences, 18(5), 927-934.
Yu, R., & Choi, Y. (2022). OkeyDoggy3D: A Mobile Application for Recognizing Stress-Related Behaviors in Companion Dogs Based on Three-Dimensional Pose Estimation through Deep Learning. Applied Sciences, 12(16), 8057.
Johnson, D. C., & Chander, M. (2021). Demographic and perception studies of a mobile application among dog breeders and owners. Journal of Extension Education, 33(2), 6653-6661.
L98 - RE app design, may be worth stating whether it is a free app? How has it been designed (ie via consultation with subject specialists?)? Is it accredited?
“Whilst an exhaustive breakdown of the app content is beyond the scope of this introduction” – can you add this in supplementary material for a record / to allow comparison with other apps and their evaluation in future. May assist understanding of “chapters” referred to in methods too.
Methods
Add survey launch date to line L118 / L138
Table 1 should be in results section - needs to explain data presented (e.g. presumably figures in brackets are percentages?)
Nice use of OSF
How was model fit evaluated? Suggest including a list of key predictors within brackets in text to save having to visit Supp. Material for this. Not clear what 1|2, 2|3, 3|4 are in predictors list, presume all predictors are related to dogs and not owners 😉
Table 3 may be more concise/easily readable as a figure? Not sure of its overall value to this specific conversation – may be better in supp material.
Results
Ensure only results are presented in this section, any interpretation should be left for discussion (see section 3.2)
Is it worth presenting an overview of patterns in other model variables too? The survey collected a lot of data - simply looking at trends in dogs and their owners who choose to use the app could be valuable to research in this area and targeting future users/ dog guardians – can you add this to the discussion too (comparatively to other training platforms?)
L237-233 – More suitable for discussion section?
L245-251 is describing what figure 2 shows so not needed - could be summarised more broadly as “the probabilities for the “No problem at all” category increased with percentage completion for the behaviours X, Y, Z”
Figure 2 – nice figure
L255-258 – can you make this clearer? Some of phrasing in results section could be made clearer / more concise – check throughout section.
Discussion
Worth a discussion of population demographics and any surprises or common patterns there (in dog ownership, training uptake, app use etc based on other published lit) – any limitations or bonuses to app use that other training platforms tend not to capture?
Plans to expand study? E.g. PBs and time left at home – links to completion likelihood?
L370 – Would be interested to see this expanded to put the results into greater context. Overall, results suggest a benefit to use for certain behaviours so would be interested to hear authors thoughts on why this is – is it that these behaviours are more responsive to training solutions, is it the specific advice the app provided or how it provided it? (as mentioned in intro some more detail on this would help determine what about “apps” are better or different to other learning platforms).
L301 – worth discussing why this may be the case re the types of behaviour, causes and training to resolve.
What advice does the app give in terms to key PBs and is it known whether this is effective when presented in other formats to dog caregivers too? (i.e. is there any way to tease apart the access to info from using the app specifically)
May be of value to include the point that those attracted to using an app may be so because find learning from one easier. Here some discussion of apps generally as learning tools – and pros and cons may be of interest to reader.
Conclusion
Include future research recommendations or are future studies with this data planned? Lots of potential here.
Supp. Material
Worth adding final column total to table S1?
In model results tables under predictors what is 1|2, 2|3, 3|4?
Reviewer 3 Report
Comments and Suggestions for Authors
This review is very well-written, which is refreshing, and the subject matter is interesting and timely. Though its concept is unique, I have concerns:
1. It was carried out by the creators of the app in question, yet I did not see any verbiage stating how objectivity was independently maintained. It would have been better carried out by an independent researcher;
2. I am skeptical about the purely inferential data, as interpreted by the authors. In the questionnaire, behaviors were not described. ""Have an issue" (the dog...or the owner) with was the only phrase used in the questionnaire. This unfortunately may have left too much to interpretation: one owner might "have an issue with" the intensity of a mouthing behavior, for example, while another may find a mild bite not an "issue." In this way, it is hard to know if we are comparing behaviors across subjects, both owners and dogs.
Round 2
Reviewer 3 Report
Comments and Suggestions for Authors
I applaud the work, and this is very well-written, good job. However, I do not see my primary objection addressed, that of the fact the app developers are presenting the research. Objectivity is the cornerstone of good science, and if an outside company did this research, it would have a more robust impression. It is still interesting, it just does not meet the standard of objective science.
Also, as a behavior analyst, I would want to see descriptive language of behaviors, rather than "have an issue" with and those types of phrases. Also some single-subject data would have been helpful.